# PREDICTIVE CVaR Q-LEARNING

**Ju-Hyun Kim**
Advanced Instrumentation Institute
Korea Research Institute of Standards and Science (KRISS)
Yuseong-gu, Daejeon 34113, Republic of Korea
jhkim@kriss.re.kr

**Seungki Min**
Business School
Seoul National University
Dongjak-gu, Seoul 08826, Republic of Korea
mskyt@sun.ac.kr

## ABSTRACT

We propose a sample-efficient Q-learning algorithm for reinforcement learning with the Conditional Value-at-Risk (CVaR) objective. Our method introduces two key innovations. First, we propose the *predictive tail value function*, a novel formulation of risk-sensitive action value, admits a recursive structure as in the conventional risk-neutral Bellman equation. This novel formulation addresses the problem of noisy policy evaluation originating from the non-decomposable objective. Second, we introduce a *two-way exploration* strategy that explores the agent's risk-sensitivity level in addition to its actions. This technique mitigates the "blindness to success" phenomenon by preventing premature convergence to overly conservative policies. We establish a rigorous theoretical foundation for this framework, including a new Bellman optimality equation and a policy improvement theorem. Empirical results demonstrate that our algorithm significantly improves both CVaR performance and learning stability.

## 1 INTRODUCTION

In high-stakes sequential decision-making tasks, the consequences of rare but catastrophic outcomes cannot be ignored. Standard reinforcement learning (RL), which assumes risk neutrality and optimizes expected returns, is insufficient in such settings. Indeed, risk-sensitive approaches have demonstrated superior performance in a variety of safety-critical domains, such as autonomous driving (Wen et al., 2020), robotic surgery (Pore et al., 2021), and finance (Greenberg et al., 2022). Among the various risk measures, Conditional Value-at-Risk (CVaR) has emerged as a prominent objective, valued for its mathematical tractability (Rockafellar et al., 2000) and its direct focus on worst-case outcomes.

CVaR quantifies the expected loss within the worst-case quantile of a return distribution, making it a natural fit for agents designed to be averse to catastrophic losses. Mirroring the development of risk-neutral RL, methodological advancements to optimize the CVaR objective have primarily evolved along two main avenues: policy-gradient methods (Tamar et al., 2015; Rajeswaran et al., 2016; Tamar et al., 2016; Queeney et al., 2021; Urpí et al., 2021; Greenberg et al., 2022; Markowitz et al., 2023; Kim & Min, 2024; Mead et al., 2025) and value-based approaches (Bäuerle & Ott, 2011; Chow et al., 2015; Pflug & Pichler, 2016; Stanko & Macek, 2019; Singh et al., 2020; Zhang & Weng, 2021; Lim & Malik, 2022; Li et al., 2022; Wang et al., 2023). Despite this progress, applying these methods introduces significant challenges.

CVaR RL is notoriously sample-inefficient, a problem often attributed to its focus on a narrow subset of worst-case trajectories. In this work, we argue that this inefficiency stems from two more fundamental issues: **noisy policy evaluation** due to a lack of temporal decomposition, and **ineffective exploration** caused by ignoring successful outcomes.

The first fundamental issue stems from the difficulty of temporal credit assignment. In many value-based formulations, the CVaR objective is treated as a single, non-decomposable reward realized only at the end of the episode. This structure prevents the agent from assessing the immediate impact of its actions, as the learning signal from an entire trajectory is collapsed into one delayed outcome. This lack of temporal decomposition makes policy evaluation exceptionally noisy, which is a primary driver of the sample inefficiency observed in CVaR RL (Hau et al., 2023; Kim & Min, 2024).

Compounding this evaluation problem is a fundamental difficulty with exploration. Because the CVaR objective is driven by the worst-case outcomes, the learning signal is derived almost exclusively from "failure" trajectories. High-return trajectories that fall outside the lower-risk quantile are effectively ignored, preventing the agent from learning to improve upon already successful behaviors. This well-documented phenomenon, known as "blindness to success" (Greenberg et al., 2022; Mead et al., 2025), can cause the learning process to stagnate in overly conservative, suboptimal policies.

To address these challenges, we introduce **Predictive CVaR Q-learning**, a novel value-based algorithm built upon a new, recursive formulation of the CVaR objective. We provide a rigorous theoretical foundation for our method, justifying each component of the algorithm, and demonstrate its superior performance and sample efficiency through experiments.

Our primary contribution is a new pair of value functions—the **predictive tail value function** and the **predictive tail probability function**—that resolves the temporal credit assignment problem. This approach adapts and extends an idea originally proposed in the policy-gradient setting (Kim & Min, 2024) for value-based learning. Together, these functions allow us to reformulate the CVaR objective into a temporally decomposable structure. We prove that this formulation satisfies a risk-neutral Bellman-style recursion, allowing the learning signal to be propagated at every step of a trajectory. This provides dense, immediate feedback for policy evaluation, drastically reducing estimation noise and improving sample efficiency.

Our second contribution is a **two-way randomized exploration** strategy designed to mitigate "blindness to success." In addition to conventional $\epsilon$-greedy for action-level exploration, we introduce novel exploration in the augmented state space by randomizing the initial risk budget. This encourages the agent to experience trajectories with varying degrees of risk sensitivity—sometimes acting boldly, other times conservatively. This exploration of risk preferences prevents the agent from prematurely converging to overly safe, suboptimal policies and promotes the discovery of more robust strategies.

## 2 PROBLEM SETUP AND PRELIMINARIES

We consider a finite-horizon Markov decision process (MDP) $\mathcal{M} = \left(\mathcal{S}, \mathcal{A}, T, (\mathcal{P}_t)_{t=1}^T, s_1\right)$, where $\mathcal{S}$ is the state space, $\mathcal{A}$ is the action space, $T$ is the time horizon, $s_1 \in \mathcal{S}$ is the initial state, and $\mathcal{P}_t$ is the transition kernel at time $t$. At each time step $t = 1, \ldots, T$, the agent observes the current state $S_t \in \mathcal{S}$, chooses an action $A_t \in \mathcal{A}$, receives a reward $R_t \in \mathbb{R}$, and transitions to a next state $S_{t+1} \in \mathcal{S}$ according to the transition kernel $\mathcal{P}_t$, i.e., $(R_t, S_{t+1}) \sim \mathcal{P}_t(\cdot \mid S_t, A_t)$.

For notational convenience, we define $R_{s:t} := \sum_{\tau=s}^t R_\tau$, and $(x)^+ := \max\{x, 0\}$.

**CVaR optimization**   Given a risk level $q \in (0, 1]$, the CVaR of a random variable $X$ is defined as

$$\text{CVaR}_q[X] := \frac{1}{q} \int_0^q \text{VaR}_u[X] du,$$

where $\text{VaR}_q[X] := \sup\{\eta \in \mathbb{R} | \mathbb{P}(X \leq \eta) \leq q\}$ denotes the Value-at-Risk at the risk level $q$, i.e., the $q$-quantile of the distribution of total reward.

Our goal is to find the optimal policy that maximizes the CVaR value of the total reward $R_{1:T}$ at the risk level $q$:

$$\sup_{\pi \in \Pi} \left\{\text{CVaR}_q^\pi [R_{1:T}]\right\},$$

where $\Pi$ is the set of non-anticipating policies, including randomized ones. More formally, let $H_t := (S_1, A_1, R_1, \ldots, S_{t-1}, A_{t-1}, R_{t-1}, S_t)$ be the history revealed up to time $t$, and let $\mathcal{H}_t$ be its

space. Each policy $\pi \in \Pi$ is defined as a sequence of functions $(\pi_t : \mathcal{H}_t \to \Delta^{|\mathcal{A}|})_{t=1}^T$ such that each $\pi_t$ specifies the distribution over actions at time $t$ given history, i.e., $A_t \sim \pi_t(\cdot \mid H_t)$.

**State space augmentation** As one of its most favored properties, the CVaR measure admits a variational representation that provides a more tractable alternative to its original definition that involves non-smooth and non-linear structure. Specifically, for any non-anticipating policy $\pi \in \Pi$, the CVaR can be expressed in the following variational form:

$$q \cdot \mathrm{CVaR}_q^\pi[R_{1:T}] = \max_{\eta \in \mathbb{R}} \left\{ q\eta + \mathbb{E}^\pi \left[ -(\eta - R_{1:T})^+ \right] \right\}.$$

Here, the factor $q$ on the left hand side is introduced to simplify expressions in later parts. This variational form allows us to reinterpret CVaR maximization as a two-stage optimization with respect to the tail risk budget $\eta$ (outer) and the policy $\pi$ (inner):

$$\sup_{\pi \in \Pi} \left\{ q \cdot \mathrm{CVaR}_q^\pi[R_{1:T}] \right\} = \max_{\eta \in \mathbb{R}} \left\{ q\eta + \sup_{\pi \in \Pi} \mathbb{E}^\pi \left[ -(\eta - R_{1:T})^+ \right] \right\}.$$

This representation naturally leads to the idea of state space augmentation, with an additional state variable representing the tail budget. Formally, we introduce a residual tail budget process $(Y_t^\eta)_{t=1}^{T+1}$ defined as

$$Y_t^\eta := \eta - R_{1:t-1},$$

where $\eta \in \mathbb{R}$ is an auxiliary variable specifying the (initial) tail budget.

A Markov policy living on the augmented state space chooses the current action $A_t$ based on the current state $S_t$ and the current residual budget $Y_t^\eta$. Such a policy is specified by an augmented Markov policy *kernel* $\chi$ that is defined as a sequence of functions $(\chi_t)_{t=1}^T$ such that $\chi_t : \mathcal{S} \times \mathbb{R} \to \Delta^{|\mathcal{A}|}$ maps the current state and the residual budget to an action distribution. That is, under a policy induced by a kernel $\chi$ with a tail budget $\eta$, the action $A_t$ is chosen according to

$$A_t \sim \chi_t(\cdot \mid S_t, Y_t^\eta).$$

Note that a kernel $\chi$ alone does not define a non-anticipating policy $\pi \in \Pi$; it must be coupled with a specific tail budget $\eta$ and we write $\mathbb{P}^{\chi,\eta}$ to denote their corresponding probability measure. We denote by $\mathcal{X}$ the set of augmented Markov policy kernels.

Notably, Bäuerle & Ott (2011) show that, for dynamic CVaR optimization, it is sufficient to search over the augmented Markov policies instead of the entire set of non-anticipating policies:

$$\sup_{\pi \in \Pi} \left\{ q \cdot \mathrm{CVaR}_q^\pi[R_{1:T}] \right\} = \max_{\eta \in \mathbb{R}} \left\{ q\eta + \sup_{\chi \in \mathcal{X}} \mathbb{E}^{\chi,\eta} \left[ -(\eta - R_{1:T})^+ \right] \right\}. \tag{1}$$

A line of work (Bäuerle & Ott, 2011; Pflug & Pichler, 2016; Wang et al., 2023) applies dynamic programming (DP) principles based on this observation. Our approach also builds on this insight, but adopts a different formulation, as detailed below.

**Dynamic programming on augmented state space** Viewed as a risk-neutral optimization, the inner optimization, $\sup_{\chi \in \mathcal{X}} \mathbb{E}^{\chi,\eta} \left[ -(\eta - R_{1:T})^+ \right]$, can be solved by applying the DP principles on the augmented state space (Pflug & Pichler, 2016).

In particular, aforementioned prior studies postulate a risk-neutral decision maker who receives a total reward of $-(\eta - R_{1:T})^+$ at the end of horizon. They introduce the action value function of an augmented Markov policy kernel $\chi$ as

$$u_t^\chi(s, y, a) := \mathbb{E}^{\chi,\eta} \left[ -(\eta - R_{1:T})^+ \mid S_t = s, Y_t^\eta = y, A_t = a \right],$$

which leads to the following Bellman equation:

$$u_t^\chi(s, y, a) = \mathbb{E}_{(R_t, S_{t+1}) \sim \mathcal{P}_t(\cdot \mid s, a), A_{t+1} \sim \chi_{t+1}(\cdot \mid S_{t+1}, y - R_t)}[u_{t+1}^\chi(S_{t+1}, y - R_t, A_{t+1})], \tag{2}$$

with $u_{T+1}^\chi(s, y, a) = -(y)^+$. Conventional risk-neutral Q-learning algorithms can be applied to optimize the kernel $\chi$ so that the optimal action value function $u_t^*$ can be obtained, and then the outer optimization reduces to a simple one-dimensional optimization, $\max_{\eta \in \mathbb{R}} \{q\eta + \max_a u_1^*(s_1, \eta, a)\}$.

However, this approach suffers from sample inefficiency. Among the sample trajectories collected over the course of Q-learning procedure, only a small subset of them will be meaningfully utilized since the effective reward will be zero, i.e., $(\eta - R_{1:T})^+ = 0$, in the majority of trajectories (roughly, $(1-q)$-fraction of trajectories). The main issue is that the term $(\eta - R_{1:T})^+$ is non-separable across time steps, effectively delaying the reward realizations to the end of time horizon.

## 3 THEORETICAL FOUNDATIONS

Existing CVaR-based dynamic programming approaches often rely on augmented state representations and treat the CVaR term as a terminal, non-decomposable objective. This leads to significant sample inefficiency and complicates recursive value estimation. In this work, we resolve this by introducing a novel, recursive formulation for the CVaR objective. Our key idea is to define a pair of predictive functions — the predictive tail value function and the predictive tail probability function — that permit temporal decomposition and satisfy a risk-neutral Bellman-style recursion. This structure facilitates value propagation and policy improvement in a manner analogous to standard Q-learning, forming the theoretical bedrock for our sample-efficient Predictive CVaR Q-learning algorithm.

**Definition 1** (Predictive tail value/probability functions). *Given an augmented Markov policy kernel $\chi$, its predictive tail value function $f^\chi = (f_t^\chi : \mathcal{S} \times \mathbb{R} \times \mathcal{A} \to \mathbb{R})_{t=1}^{T+1}$ is defined as*

$$f_t^\chi(s, y, a) := \mathbb{E}^{\chi, \eta=0} \left[ \mathbb{I}\{R_{t:T} \leq y\} R_{t:T} \mid S_t = s, Y_t^{\eta=0} = y, A_t = a \right],$$

*with $f_{T+1}^\chi(s, y, a) := 0$. Additionally, its predictive tail probability function $g^\chi = (g_t^\chi : \mathcal{S} \times \mathbb{R} \times \mathcal{A} \to [0, 1])_{t=1}^{T+1}$ is defined as*

$$g_t^\chi(s, y, a) := \mathbb{P}^{\chi, \eta=0} \left( R_{t:T} \leq y \mid S_t = s, Y_t^{\eta=0} = y, A_t = a \right), \tag{3}$$

*with $g_{T+1}^\chi(s, y, a) := \mathbb{I}\{0 \leq y\}$.*

Here, the choice $\eta = 0$ is arbitrary. Above notion of predictive tail values and probabilities are invariant in $\eta$.

The *predictive tail probability* function $g_t^\chi$ quantifies the likelihood that the remaining return from time $t$ onward falls below a specified threshold given the current state, residual budget, and action. This object effectively encodes the probability of entering the CVaR tail conditioned on the current decision point. The idea of modeling such risk-conditioned probabilities is reminiscent of the Predictive CVaR Policy Gradient framework (Kim & Min, 2024), which uses similar quantities to inform policy gradient-based policy updates. In contrast, we integrate action-conditioning into the predictive structure and embed it within a Bellman-style recursion. This formulation enables value-based approach such as Q-learning, moving beyond trajectory-level estimation and allowing for action selection that directly maximize the CVaR objective.

In addition to the predictive tail probability, we define the *predictive tail value* function — formally described by the function $f_t^\chi(s, y, a)$ — as a novel, risk-sensitive analogue of the standard action value function (so called Q-function). This function captures the expected cumulative return weighted by the probability that the trajectory remains in the CVaR tail from time $t$ onward. This function reflects both the magnitude and likelihood of tail outcomes. Importantly, the return $R_{t:T}$ is recursively decomposable, which allows $f^\chi$ to satisfy a risk-neutral Bellman-style recursion. This recursive structure enables value propagation and policy improvement analogous to standard Q-learning, while preserving sensitivity to risk throughout the learning process.

Next result shows that the objective of the inner optimization, $\mathbb{E}^{\chi, \eta} \left[ -(\eta - R_{1:T})^+ \right]$, can be represented in terms of $f^\chi$ and $g^\chi$, and also that $f^\chi$ can be decomposed across time steps.

**Assumption 1.** *Under any non-anticipating policy $\pi \in \Pi$ and any time $t \in \{1, \ldots, T\}$, the distribution of remaining return $R_{t:T}$ does not have any probability mass.*

**Proposition 1** (Temporal decomposition). *Under Assumption 1, for any $s \in \mathcal{S}, y \in \mathbb{R}, a \in \mathcal{A}$, and $t \in \{1, \ldots, T\}$, the following equations hold:*

$$f_t^\chi(s, y, a) = \mathbb{E}^{\chi,\eta}\left[\left.\sum_{\tau=t}^{T} g_{\tau+1}^\chi(S_{\tau+1}, Y_{\tau+1}, A_{\tau+1}) \times R_\tau \;\right|\; S_t = s, Y_t^\eta = y, A_t = a\right],$$

$$g_t^\chi(s, y, a) = \mathbb{E}^{\chi,\eta}\left[\left. g_{t+1}^\chi(S_{t+1}, Y_{t+1}, A_{t+1}) \;\right|\; S_t = s, Y_t^\eta = y, A_t = a\right], \tag{4}$$

*and*

$$\mathbb{E}^{\chi,\eta}\left[-(\eta - R_{1:T})^+\right] = \mathbb{E}_{A_1 \sim \chi_1(\cdot|s_1,\eta)}\left[f_1^\chi(s_1, \eta, A_1) - g_1^\chi(s_1, \eta, A_1) \times \eta\right].$$

We next show that the predictive tail value function exhibits a recursive structure that is very analogous to the standard Bellman equation in the risk-neutral setting.

**Theorem 1** (Bellman equation). *Given an augmented Markov policy kernel $\chi$, under Assumption 1, its predictive tail value function $f^\chi$ and predictive tail probability function $g^\chi$ satisfy*

$$f_t^\chi(s, y, a) = \mathbb{E}_{(R_t, S_{t+1}) \sim \mathcal{P}_t(\cdot|s,a), A_{t+1} \sim \chi_{t+1}(\cdot|S_{t+1}, y-R_t)}\left[f_{t+1}^\chi\big(S_{t+1}, y - R_t, A_{t+1}\big)\right.$$
$$\left. + g_{t+1}^\chi\big(S_{t+1}, y - R_t, A_{t+1}\big) \times R_t\right], \tag{5}$$

*for all $s \in \mathcal{S}$, $y \in \mathbb{R}$, $a \in \mathcal{A}$, and $t \in \{1, \ldots, T\}$.*

Compared to the Bellman equation (Eq. 2) established in prior work, this Bellman equation (Eq. 5) includes an additional term, $g_{t+1}^\chi\big(S_{t+1}, y - R_t, A_{t+1}\big) \times R_t$, corresponding to the immediate reward in the standard Bellman equation. This term reflects that the anticipated contribution of the current reward to the objective, enabling efficient value propagation in our suggested CVaR Q-learning algorithm.

Building on this recursion, we can derive the optimality conditions that the optimal kernel has to satisfy and its implication on the CVaR objective. This allows us to characterize optimality and construct improvement principles, despite the non-Markovian nature of CVaR. We begin by defining the notion of greedy kernels for the state space augmentation, mirroring the classic notion of greedy policies in the standard Q-learning approaches.

**Definition 2** (Greedy kernel). *An augmented Markov policy kernel $\chi$ is said to be greedy with respect to a predictive tail value/probability function pair $(f, g)$ if*

$$\{a \in \mathcal{A} \mid \chi_t(a|s, y) > 0\} \subseteq \underset{a \in \mathcal{A}}{\arg\max}\left\{f_t(s, y, a) - g_t(s, y, a) \times y\right\},$$

*for all $s \in \mathcal{S}$, $y \in \mathbb{R}$, and $t \in \{1, \ldots, T\}$.*

**Theorem 2** (Bellman optimality equation). *Define*

$$v_t^\chi(s, y) := \mathbb{E}_{A_t \sim \chi_t(\cdot|s,y)}\left[f_t^\chi(s, y, A_t) - g_t^\chi(s, y, A_t) \times y\right], \quad v_t^*(s, y) := \sup_{\chi \in \mathcal{X}} v_t^\chi(s, y).$$

*Then, the following holds under Assumption 1:*

1. *Let $\Pi(\chi)$ be the set of augmented Markov policies induced by a kernel $\chi$ across all values of $\eta \in \mathbb{R}$. Then,*

$$\sup_{\pi \in \Pi(\chi)}\left\{q \cdot CVaR_q^\pi[R_{1:T}]\right\} = \max_{\eta \in \mathbb{R}}\left\{q\eta + v_1^\chi(s_1, \eta)\right\}.$$

2. *With respect to all non-anticipating policies,*

$$\sup_{\pi \in \Pi}\left\{q \cdot CVaR_q^\pi[R_{1:T}]\right\} = \max_{\eta \in \mathbb{R}}\left\{q\eta + v_1^*(s_1, \eta)\right\}.$$

3. *$v^\chi \equiv v^*$ if and only if $\chi$ is greedy with respect to $(f^\chi, g^\chi)$.*

The functions $v_t^\chi$ and $v_t^*$ are analogous to the state value function and the optimal state value function, respectively, in the risk-neutral setting. The above result establishes explicit connections between the CVaR objective and predictive tail value/probability functions, and clarifies the meaning of optimizing the augmented Markov policy kernel $\chi$ through dynamic programming principles.

We now demonstrate that the state-wise improvement of the kernel (in the augmented state space) indeed improves the CVaR performance of the resulting policies.

**Theorem 3** (Policy improvement). *Consider an augmented Markov policy kernel $\chi$ along with its predictive tail value function $f^\chi$ and predictive tail probability function $g^\chi$. Let $\chi'$ be the greedy kernel with respect to $(f^\chi, g^\chi)$. Then, under Assumption 1,*

$$v_t^\chi(s, y) \leq v_t^{\chi'}(s, y), \quad \forall s \in \mathcal{S}, y \in \mathbb{R}, t \in \{1, \ldots, T\}.$$

*Consequently,*

$$\sup_{\pi \in \Pi(\chi)} CVaR_q^\pi[R_{1:T}] \leq \sup_{\pi \in \Pi(\chi')} CVaR_q^\pi[R_{1:T}], \tag{6}$$

*for any $q \in (0, 1]$.*

This result formally guarantees that taking greedy updates with respect to the predictive functions yields monotonic improvement in the CVaR objective over the entire augmented state space. This result generalizes the classical policy improvement theorem to the risk-sensitive setting and justifies our value-based approach to CVaR optimization.

In addition, estimating tail probability and tail value separately offers an important advantage rather than estimating a tail expectation itself. In particular, this advantage becomes most pronounced in the challenging cases (1) where the number of trajectories with non-zero effective reward is small, and (2) where the rewards beyond the threshold exhibit large variance. The tail probability can still be estimated in a relatively stable way—e.g., by counting the proportion of trajectories exceeding the threshold. In contrast, estimating a tail expectation requires regressing on the magnitude of the tail values, which becomes considerably more unstable when the effective-reward samples are both few and high-variance. Our method explicitly separates probability estimation (via $(g^\chi)$) and value estimation (via $(f^\chi)$) through the temporal decomposition we introduce. This decoupling allows the probability component to be learned in a more stable manner, while the value component can leverage the decomposition structure to reduce variance. Moreover, since $g^\chi$ represents a probability bounded in ([0,1]), it can benefit from stable learning strategies, as the reviewer noted, such as log-loss or KL-divergence based objectives, further mitigating noise.

## 4 ALGORITHM

Based on the theoretical framework established above, we now present our Predictive CVaR Q-learning (`PCVaR-Q`) algorithm. The algorithm learns three key components: two function approximators, $\hat{f}^\theta$ and $\hat{g}^\phi$, which estimate the predictive tail value and probability, and a risk-budget parameter $\eta$, which tracks the optimal risk level. The learning process is driven by a two-way exploration strategy and periodic updates to these parameters, as detailed in Algorithm 1.

**Generating sample trajectories with two-way randomized exploration** To generate sample trajectories, the agent follows a behavior policy built upon a two-way randomized exploration strategy (lines 4 – 9). The first component is an $\epsilon$-greedy scheme for action-level exploration. This applies random perturbations to the actions of the greedy policy derived from the current function approximators, $\hat{f}^\theta$ and $\hat{g}^\phi$.

The second, more distinctive component of this strategy is the exploration within the augmented state space. In line 4, the initial residual budget $Y_1$ is sampled from a normal distribution $\mathcal{N}(\eta, \sigma_k^2)$ instead of being fixed to a single value. Here, $\eta$ corresponds to current best estimate of the optimal risk budget. By sampling around this central value, the agent is prompted to experience trajectories under varying degrees of risk sensitivity, thereby exploring the space of risk preferences more effectively. Analogous to $\epsilon_k$ for action-level exploration, the variance $\sigma_k^2$ governs the extent of this exploration and is typically annealed as training progresses.

Our approach follows this established practice: risk-level exploration provides a flexible heuristic whose hyperparameters can be tuned through standard experimental procedures, just as is common for classical $\epsilon$-greedy exploration strategies. Similar to $\epsilon$-greedy exploration, its exploration is governed by two design choices: the initial value of $\sigma$ and the schedule by which $\sigma$ decays over time. The initial $\sigma$ can be reasonably specified from prior domain knowledge or from basic properties of the environment. The decay schedule, much like the choice of $\epsilon$-decay in $\epsilon$-greedy, admits multiple reasonable forms and is ultimately selected based on empirical validation.

---

**Algorithm 1** Predictive CVaR Q-Learning Algorithm (`PCVaR-Q`)

---

1: **Initialize** parameters $(\theta, \phi, \eta)$, risk budget grid $H$, replay buffer $\mathcal{D}$.
2: **Pretrain** parameters $(\theta, \phi, \eta)$ using pre-existing sample trajectories, if available.
3: **for** $k = 1$ to $K$ **do**
4:     Set $S_1 \leftarrow s_1$, and sample $Y_1 \sim \mathcal{N}(\eta, \sigma_k^2)$.
5:     **for** $t = 1$ to $T$ **do**
6:         With probability $\epsilon_k$, choose $A_t \sim \text{Uniform}(\mathcal{A})$.
7:         Otherwise, choose $A_t$ greedily with respect to $(\hat{f}^\theta, \hat{g}^\phi)$:

$$A_t \leftarrow \arg\max_{a \in \mathcal{A}} \left\{ \hat{f}_t^\theta(S_t, Y_t, a) - \hat{g}_t^\phi(S_t, Y_t, a) \cdot Y_t \right\}.$$

8:         Execute $A_t$, observe $R_t, S_{t+1}$; update $Y_{t+1} \leftarrow Y_t - R_t$.
9:         Store $(R_{1:t-1}, S_t, A_t, R_t, S_{t+1})$ into replay buffer $\mathcal{D}$.
10:        Sample a mini-batch $\mathcal{B} = \{(R_{1:j-1}, S_j, A_j, R_j, S_{j+1})\}_{j=1}^B$ from $\mathcal{D}$.
11:       **Update** $\theta$ and $\phi$ using TD losses, Eq. 7 and Eq. 8:

$$\theta \leftarrow \theta - \alpha_{\theta,k} \nabla_\theta \mathcal{L}_f(\theta; \mathcal{B}), \quad \phi \leftarrow \phi - \alpha_{\phi,k} \nabla_\phi \mathcal{L}_g(\phi; \mathcal{B}).$$

12:     **end for**
13:     Every $c$ episodes, **update** risk budget $\eta$:

$$\eta \leftarrow \arg\max_{\eta' \in H} \max_{a \in \mathcal{A}} \left\{ \hat{f}_1^\theta(s_1, \eta', a) + \eta' \cdot \left( q - \hat{g}_1^\phi(s_1, \eta', a) \right) \right\}.$$

14: **end for**

---

**Updating the predictive functions and risk budget**   The learning process involves updating the parameters of the function approximators, denoted by $\theta$ and $\phi$ for $\hat{f}^\theta$ and $\hat{g}^\theta$ respectively, along with the risk budget parameter $\eta$ (lines 11–13). From the sampled trajectories, $\theta$ and $\phi$ are updated using two distinct temporal-difference (TD) loss functions.

The parameter $\theta$ is updated updated by minimizing the following TD loss, $\mathcal{L}_f(\theta; \mathcal{B})$, which is derived from the Bellman equation (Eq. 5) in Theorem 1:

$$\mathcal{L}_f(\theta; \mathcal{B}) := \frac{1}{B|H|} \sum_{\eta' \in H} \sum_{j=1}^B \left( \hat{f}_j^\theta(S_j, \eta' - R_{1:j-1}, A_j) \right. \tag{7}$$

$$\left. - \left[ \hat{f}_{j+1}^\theta(S_{j+1}, \eta' - R_{1:j}, A'_{j+1}) + \hat{g}_{j+1}^\phi(S_{j+1}, \eta' - R_{1:j}, A'_{j+1}) \cdot R_j \right] \right)^2$$

where $A'_{j+1}$ is the greedy action at $(S_{j+1}, \eta' - R_{1:j})$ with respect to $(\hat{f}^\theta, \hat{g}^\phi)$. A key aspect of our method is that the loss for each sample is computed over a discrete set of candidate risk budgets $H \subset \mathbb{R}$, leveraging the property that the predictive functions are invariant to the initial risk budget level. This encourages the function approximators to generalize across various risk levels.

Similarly, the parameter $\phi$ is updated by minimizing a second TD loss, $\mathcal{L}_g(\phi; \mathcal{B})$, derived from the martingale property of the predictive tail probability function (Eq. 4) in Proposition 1:

$$\mathcal{L}_g(\phi; \mathcal{B}) := \frac{1}{B|H|} \sum_{\eta' \in H} \sum_{j=1}^B \left( \hat{g}_{j+1}^\phi(S_{j+1}, \eta' - R_{1:j}, A'_{j+1}) - \hat{g}_j^\phi(S_j, \eta' - R_{1:j-1}, A_j) \right)^2. \tag{8}$$

Finally, the risk level $\eta$ is updated periodically for stability by solving the outer optimization problem in the variational formulation of CVaR (cf. Theorem 2). This newly identified optimal risk budget then serves as the central point for the agent's risk-level exploration in subsequent episodes.

This entire learning procedure constitutes a form of generalized policy iteration (GPI). The TD updates for $\theta$ and $\phi$ act as the kernel evaluation step, driving the learned functions to satisfy the Bellman optimality equation (Theorem 2). The greedy kernel derived from these updated functions is then guaranteed to be superior by Theorem 3. This interplay between evaluation and improvement ensures that our algorithm progressively finds a better kernel, converging towards a CVaR-optimal solution.

**Starting with pretrained parameters (optional)** While our core `PCVaR-Q` algorithm is inherently more robust to the "blindness-to-success" phenomenon than prior methods because it utilizes two-way exploration, this issue can still be a challenge in the early stages of learning. To further mitigate this risk, we propose an optional warm-start by pre-training the parameters $(\theta, \phi, \eta)$ (line 2). This is a highly practical step, as it can leverage any dataset of pre-existing sample trajectories (e.g., ones obtained from a risk-neutral policy). The pre-training procedure itself is straightforward, as it reuses the update rules to fit the predictive functions to this data, from which an initial risk budget $\eta$ can also be derived. This initialization anchors the agent to promising, high-return behaviors from the outset, ensuring a more stable learning dynamic.

## 5 EXPERIMENTS

In this section, we evaluate the performance of Predictive CVaR Q-learning algorithm in a controlled setting to investigate (1) whether our method improves CVaR performance over risk-neutral policies, and (2) whether it achieves stable and sample-efficient learning compared to baselines. We compare our proposed algorithm `PCVaR-Q` with the other two baselines, `RN` and `CVaR-Q`:

- `RN`: Risk-neutral optimal policy learned through risk-neutral Q-learning.
- `CVaR-Q`: The policy learned through a Q-learning-style approach based on the Bellman operator in (Pflug & Pichler, 2016), as detailed in Algorithm 2.
- `PCVaR-Q`: The policy learned through the Predictive CVaR Q-learning algorithm.

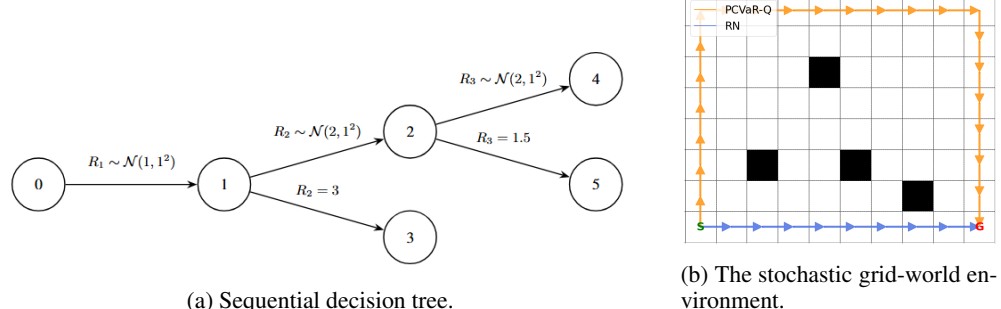

(a) Sequential decision tree.

(b) The stochastic grid-world environment.

Figure 1: Illustration of experiment environments.

### 5.1 SEQUENTIAL DECISION TREE (TREE-STRUCTURED MDP)

**Setup** We design a finite-horizon MDP to highlight risk-return trade-offs under CVaR optimization. The agent starts in State 0 and proceeds through a branching structure involving both stochastic and deterministic rewards. Figure 1a illustrates the simulated environment. The dynamics are described as follows. The agent starts in State 0. At State 0, the agent moves deterministically to State 1, receiving a stochastic reward $R_1 \sim \mathcal{N}(1, 1^2)$. At State 1 and State 2, the agent chooses between two actions, *up* and *down*. The *up* action yields stochastic rewards whereas the *down* action yields deterministic rewards and terminates the MDP process (see Figure 1a).

The total return $R_{1:3} = R_1 + R_2 + R_3$ (with missing rewards treated as zero) presents a clear risk-return trade-off: the *down* path offers lower variance and moderate return, while the *up* path provides higher expected return but greater risk. The risk-neutral optimal policy always chooses the *up* action, yielding an average total return of $5.0$ but a CVaR value of only $1.96$ at level $q = 0.1$. In contrast, the CVaR-optimal policy at the same level ($q = 0.1$) adopts a more cautious approach. A simple calculation shows that it will play *down* at State 1 if $R_1 \leq -0.2$ and will do so at State 2 if $R_1 + R_2 \leq 1.1$. The CVaR performance of this policy is $2.50$, and its corresponding VaR value is $3.02$.

**Results** To evaluate risk-sensitive behavior, we compare the distributions of total reward $R_{1:3}$ under the policies obtained by running `RN` and `PCVaR-Q` over 100,000 episodes. As shown in

Figure 2a, `PCVaR-Q` yields a higher lower-tail return, achieving a CVaR value of 2.45 at risk level $q = 0.1$, whereas `RN` achieves 1.96 (the optimal CVaR value is 2.50). This confirms that `PCVaR-Q` successfully finds out the CVaR-optimal policy.

Figure 2b tracks the evolution of CVaR performance ($q = 0.1$) throughout the training process: Each solid line represents the mean performance across 10 independent trials. Policies were evaluated every 100 iterations, with each point estimated from 100,000 sample runs. Our `PCVaR-Q` algorithm exhibits a stable learning curve that steadily converges to a near-optimal value, whereas the `CVaR-Q` baseline shows high variance and converges to a suboptimal policy. This result confirms the superior stability and sample efficiency of our method, validating the practical benefits of our proposed recursive structure in a risk-sensitive learning context.

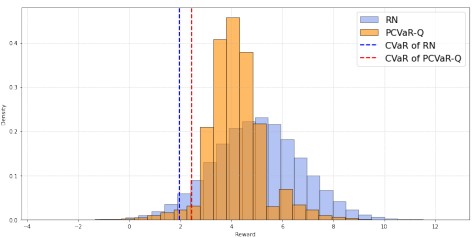 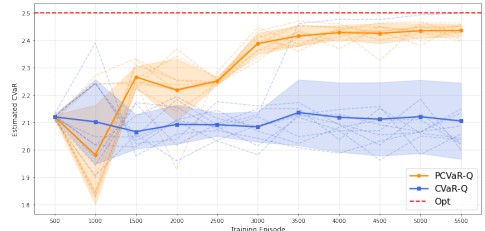

(a) Histograms of total reward for the final `PCVaR-Q` and risk-neutral (`RN`) policies.

(b) Actual CVaR performance ($q = 0.1$) over the course of Q-learning.

Figure 2: Results for sequential decision tree

## 5.2 GRID-WORLD WITH STOCHASTIC TRANSITION AND OBSTACLES.

**Setup**  We conduct our experiments in a grid-based navigation environment designed to evaluate decision-making under uncertainty and risk. The environment is a two-dimensional grid of size 8 (height) $\times$ 10 (width), where an agent starts at the bottom-left corner and aims to reach a goal located at the bottom-right corner. The agent's action space consists of four cardinal movements: *up*, *down*, *left*, and *right*. The transition is stochastic: an intended action is successful with a probability of 0.7, but with a 0.3 probability, the agent moves in one of the other three directions, chosen uniformly at random. The episode terminates if the agent reaches the goal, which yields a reward from $\mathcal{N}(50, 1)$, or hits an obstacle, which incurs a penalty from $\mathcal{N}(-50, 1)$. All other transitions receive a reward of $-1$ as a step penalty. While the risk-neutral policy tends to follow a direct path toward the goal, it often incurs a higher chance of collision with obstacles. In contrast, the risk-sensitive policy, such as PCVaR-Q, learns a safer path that avoids obstacles, even if it results in a longer trajectory. Figure 1b illustrates the experimental environment and the distinct policies learned by the risk-neutral (`RN`) and `PCVaR-Q` agents.

**Results**  Figure 3a presents the total reward distributions from 50,000 evaluation episodes, showing that PCVaR-Q produces a more robust policy. By mitigating the risk of catastrophic penalties, it improves the lower tail of the distribution, achieving a CVaR of $-55.84$ at the $q = 0.1$ level — a clear improvement over the RN policy's $-58.37$ and close to the optimal value of $-53.34$.

Furthermore, Figure 3b illustrates that this superior outcome is achieved through a more efficient and stable learning process. The CVaR performance of `PCVaR-Q`, measured every 1000 iterations, shows rapid and consistent convergence, while the baseline model is more volatile. This demonstrates that our method's improved sample efficiency directly translates to better and more reliable risk-aware policies in complex environments.

## 6 CONCLUSION

We introduced Predictive CVaR Q-learning (`PCVaR-Q`), a novel CVaR Q-learning framework designed to optimize CVaR objectives in a sample-efficient and theoretically grounded manner. Our key contributions include the predictive tail value function, which enables a recursive Bellman structure tailored to CVaR, and a two-way randomized exploration strategy that explores both action and

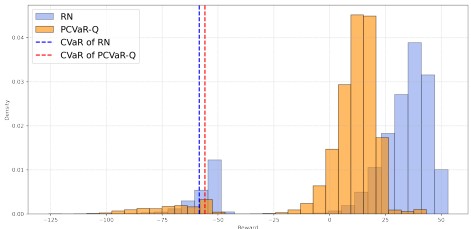
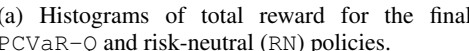
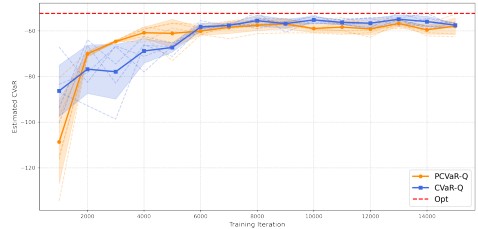

(a) Histograms of total reward for the final `PCVaR-Q` and risk-neutral (`RN`) policies.

(b) Actual CVaR performance ($q = 0.1$) over the course of Q-learning.

Figure 3: Results for stochastic grid-world.

risk budget spaces. Together, we provide theoretical guarantees including a Bellman equation, optimality condition, and policy improvement theorem specific to the CVaR setting. These results offer a principled foundation of temporal-difference (TD) learning style algorithm, thereby extending classical Q-learning theory into the domain of CVaR measure. Experimental results further demonstrate the practical effectiveness and stability of our approach.

One limitation of our method is the increased model complexity: the introduction of the predictive tail value and probability functions requires learning two separate function approximators, and the residual threshold $\eta$ must also be tracked and updated throughout the learning process. Despite this added complexity, our results show that the benefits in sample efficiency and theoretical rigor justify the overhead in most settings. Future work may extend our framework to deep RL environments, and explore integration with model-based risk-sensitive planning. Our method also opens the door for further research on adaptive risk modeling and safety-critical learning in real-world applications.

**Reproducibility Statement.** We have made significant efforts to ensure the reproducibility of our results. Baseline algorithmic descriptions and detailed experimental settings, including hyperparameters, are described in Appendix A.

### ACKNOWLEDGMENTS

This work was supported by the New Faculty Startup Fund from Seoul National University.

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

# A  EXPERIMENT AND IMPLEMENTATION DETAILS

## A.1  BASELINE: CVAR-Q ALGORITHM

We employ CVaR Q-learning to solve the CVaR optimization problem. This approach learns a function approximator $u^\theta$ that estimates the CVaR value function. The training procedure is summarized in Algorithm 2. While the method shares similarities with Predictive CVaR Q-learning, the key difference lies in the optimization of the parameter $\theta$, which is updated using a temporal-difference (TD) loss derived from the Bellman optimal equation in (Wang et al., 2023) such as: for a mini-batch $\mathcal{B} = \{(R_{1:j-1}, S_j, A_j, R_j, S_{j+1})\}_{j=1}^{B}$,

$$\mathcal{L}_u(\theta; \mathcal{B}) := \frac{1}{B|H|} \sum_{\eta' \in H} \sum_{j=1}^{B} \left( \hat{u}_j^\theta(S_j, \eta' - R_{1:j-1}, A_j) - \hat{u}_{j+1}^\theta(S_{j+1}, \eta' - R_{1:j}, A_{j+1}^*) \right] \right)^2.$$

where $A_{j+1}^* \in \arg\max_{a \in \mathcal{A}} \hat{u}_{j+1}^\theta(S_{j+1}, \eta' - R_{1:j}, a)$.

---

**Algorithm 2** CVaR Q-Learning Algorithm (`CVaR-Q`)

---

1: **Initialize** parameters $\theta$, $\phi$, risk budget grid $H$, initial $\eta \in H$, replay buffer $\mathcal{D}$
2: **for** $k = 1$ to $K$ **do**
3:     Initialize $S_1 \leftarrow s_1$, $Y_1 \leftarrow \eta$.
4:     **for** $t = 1$ to $T$ **do**
5:         With probability $\epsilon$, choose $A_t \sim \text{Uniform}(\mathcal{A})$.
6:         Otherwise, choose $A_t$ greedily with:
$$A_t \leftarrow \arg\max_{a \in \mathcal{A}} \left\{ u_t^\theta(S_t, Y_t, a) \right\}.$$
7:         Execute $A_t$, observe $R_t$, $S_{t+1}$; update $Y_{t+1} \leftarrow Y_t - R_t$.
8:     **end for**
9:     Store $(R_{1:t-1}, S_t, A_t, R_t, S_{t+1})$ into replay buffer $\mathcal{D}$.
10:     Sample a mini-batch $\mathcal{B} = \{(R_{1:j-1}, S_j, A_j, R_j, S_{j+1})\}_{j=1}^{B}$ from $\mathcal{D}$.
11:     **Update** $\theta$ using TD loss:
$$\theta \leftarrow \theta - \alpha_\theta \nabla_\theta \mathcal{L}_u(\theta; \mathcal{B}).$$
12:     **Update** risk budget $\eta$:
$$\eta \leftarrow \arg\max_{\eta' \in H} \max_{a \in \mathcal{A}} \left\{ \hat{u}_1^\theta(s_1, \eta', a) + \eta' \cdot q \right\}.$$
13: **end for**

---

## A.2 SEQUENTIAL DECISION TREE

**CVaR parameterization** To optimize for Conditional Value at Risk (CVaR), we define a discrete grid for the risk budget $H$ as follows:

$$H = \{-10.0, -9.9, -9.8, \ldots, 1.48, 14.9, 15.0\},$$

and the quantile level $q = 0.1$. The Predictive CVaR Q-learning algorithm estimates the predictive tail value function $f^\chi$ and the predictive tail probability function $g^\chi$. The CVaR Q-learning algorithm estimates value function $u^\chi$. A tabular function approximator is used for each of these functions: $f^\theta$, $g^\phi$, and $u^\theta$, where the values are maintained separately for combinations of states, residual risk budgets, and actions. The CVaR objective at a given risk budget $\eta$ is computed as:

$$\hat{v}(s_1, \eta) = \eta \cdot (q - \hat{g}^\phi(s_1, \eta, a')) + \hat{f}_1(s_1, \eta, a'),$$

or alternatively:

$$\hat{v}(s_1, \eta) = \eta \cdot q + \hat{u}_1(s_1, \eta, a'),$$

where $a' = \arg\max_{a \in \mathcal{A}} \eta \cdot (q - \hat{g}_1(s_1, \eta, a')) + \hat{f}_1(s_1, \eta, a')$ or $a' = \arg\max_{a \in \mathcal{A}} \eta \cdot q + \hat{u}_1(s_1, \eta, a')$.

**Learning and optimization** We utilize both Predictive CVaR Q-learning and CVaR Q-learning algorithms. The learning process maintains estimates of $f^\theta$, $g^\phi$, and $u^\theta$. These are updated via the Adam optimizer. Key settings include:

- Learning rates: $\alpha_\theta = 0.01$, $\alpha_\phi = 0.0001$
- Epsilon decay: $\epsilon_t = 0.1 \cdot 0.9^{\lfloor t/100 \rfloor}$
- Batch size: 8 sampled trajectories per episode
- Optimizer: Adam with $\beta_1 = 0.9$, $\beta_2 = 0.999$, $\epsilon = 10^{-8}$
- Episodes: 5,000

- Risk-level exploration scheduling:

$$\sigma_k = \begin{cases} 3, & \text{if } 0 \le k < 500 \\ 2, & \text{if } 500 \le k < 1000 \\ 1, & \text{if } 1000 \le k < 1500 \\ 0, & \text{if } k \ge 1500 \end{cases}$$

Both $f^\theta, g^\phi$ and $u^\theta$ are updated based on cumulative return trajectories. The risk level parameter $\eta$ is updated every 500 iterations. The same hyperparameter configuration is applied to both Predictive CVaR Q-learning and CVaR Q-learning algorithms to ensure consistency. Under ideal conditions, the CVaR Q-learning algorithm also converges to the optimal risk-sensitive policy. However, we aim to highlight the robustness and stability of the Predictive CVaR Q-learning algorithm under the same conditions.

## A.3 GRID-WORLD

**CVaR parameterization**   To optimize for Conditional Value at Risk (CVaR), we define a discrete grid for the risk budget $H$ as follows:

$$H = \{-150, -149, -148, \dots, 98, 99, 100\},$$

and the quantile level $q = 0.1$. A tabular function approximator is used for each of these functions: $f^\theta, g^\phi$, and $u^\theta$, where the values are maintained separately for combinations of states, residual risk budgets, and actions. The CVaR objective at a given risk budget $\eta$ is computed as same method before.

**Learning and optimization**   We utilize both Predictive CVaR Q-learning and CVaR Q-learning algorithms. The learning process maintains estimates of $f^\theta, g^\phi$, and $u^\theta$. These are updated via the Adam optimizer. Key settings include:

- Learning rates: $\alpha_\theta = 0.01$, $\alpha_\phi = 0.01$
- Epsilon decay: $\max(1 - (\frac{\text{episode}}{2000}), 0.0)$
- Batch size: 1 sample trajectories per episode
- Optimizer: fixed learning rates
- Episodes: 15,000
- Risk-level exploration scheduling:

$$\sigma_k = \begin{cases} 45, & \text{if } 0 \le k < 2000 \\ 30, & \text{if } 2000 \le k < 4000 \\ 15, & \text{if } 4000 \le k < 6000 \\ 0, & \text{if } k \ge 6000 \end{cases}$$

- Pretrain parameters $(\theta, \phi, \eta)$ with risk-neutral policy

Both $f^\theta, g^\phi$ and $u^\theta$ are updated based on cumulative return trajectories. The risk level parameter $\eta$ is updated every 500 iterations. The same hyperparameter configuration is applied to both Predictive CVaR Q-learning and CVaR Q-learning algorithms to ensure consistency.

## A.4 COMPUTATION AND IMPLEMENTATION

All experiments were conducted on a system with an 11th Gen Intel(R) Core(TM) i7-11700K processor running at 3.60 GHz, 32 GB of RAM, and 8 CPU cores. Each experiment for a given seed typically requires about 20 (tree) and 90 (grid-world) minutes to complete. The implementation is written in Python 3.8+ using only NumPy and Matplotlib, without relying on any external reinforcement learning libraries (such as Gym or Stable Baselines). All code and experimental results will be released as a ZIP file.

## B EFFECTIVENESS OF TWO-WAY EXPLORATION AND PARAMETERS PRETRAINING

In this section, we evaluate the performance of Predictive CVaR Q-learning algorithm in a controlled setting to investigate the effect of parameters pretraining and $\eta$ sampling. We compare our proposed algorithm `PCVaR-Q` with or without the two technicque, Two-way and `Pretrain`:

- `Two-way` (O/X) : exploration with both action-level and risk-level (O) or with only action-level (X).
- `Pretrain` (O/X): Start with pretrained parameters $(\theta, \phi, \eta)$ (O) or zero parameters $(\theta, \phi, \eta)$ (X)

**Result** Each row in Figure 4 visualizes the distinct paths discovered during training and the state visit frequencies as heatmaps for each model at specific iterations (left: at 2,000 iterations, right: at 12,000 iterations). In Figures 4a∼ 4d, where models were initialized with zero value functions and trained without pretraining, we observe that the agents fail to learn optimal trajectories. The case without both two-way exploration and pretraining failed to achieve sufficient exploration, resulting in poor performance. Even when two-way exploration was enabled without pretraining, the agent still converged to suboptimal local behaviors. This phenomenon reflects a blindness to success,which may bring CVaR learning to a local optimum deadlock. In contrast, models were initialized with the pretrained parameters in Figures (e)–(h) successfully identify safer trajectories. Notably, models augmented with two-way exploration converge more quickly and consistently to the desirable path, demonstrating improved exploration and stability during training.

## C PROOFS OF THEORETICAL FOUNDATIONS

To establish the theoretical foundations, we begin by introducing a key technical assumption that simplifies the treatment of conditional distributions over cumulative rewards.

**Assumption 1.** *Under any non-anticipating policy $\pi \in \Pi$ and any time $t \in \{1, \ldots, T\}$, the distribution of remaining return $R_{t:T}$ does not have any probability mass.*

Assumption 1 ensures that the cumulative return $R_{t:T}$ has a continuous distribution, with no point masses (atoms). This condition is critical to the theoretical framework presented in this work. Specifically, it guarantees that the conditional probability $\mathbb{P}(R_{t:T} \leq Y_t \mid S_t, Y_t, A_t)$ is differentiable with respect to the threshold $Y_t$, which is a key component for defining the tail probability function $g_t^\chi$ and constructing the recursive CVaR objective function. Without Assumption 1, the function $g_t^\chi$ could exhibit discontinuities or even be undefined. This is due to the presence of atoms in the conditional distribution of returns. This would invalidate several important results, including Lemma 1, Proposition 1, and Theorem 1, which rely on the smoothness of these functions.

### C.1 PROOF OF PROPOSITION 1

In order to derive efficient recursive formulations of the CVaR objective, we first characterize how the CVaR objective can be decomposed temporally. The following proposition shows that the predictive tail value function $f^\chi$ can be expressed as the expectation of cumulative discounted rewards, weighted by a predictive tail probability function $g^\chi$. This result provides the foundation for a Bellman-type recursive formulation presented in Theorem 1.

**Proposition 1** (Temporal decomposition). *Under Assumption 1, for any $s \in \mathcal{S}, y \in \mathbb{R}, a \in \mathcal{A}$, and $t \in \{1, \ldots, T\}$, the following equations hold:*

$$f_t^\chi(s, y, a) = \mathbb{E}^{\chi, \eta} \left[ \sum_{\tau=t}^{T} g_{\tau+1}^\chi(S_{\tau+1}, Y_{\tau+1}, A_{\tau+1}) \times R_\tau \,\middle|\, S_t = s, Y_t^\eta = y, A_t = a \right],$$

$$g_t^\chi(s, y, a) = \mathbb{E}^{\chi, \eta} \left[ g_{t+1}^\chi(S_{t+1}, Y_{t+1}, A_{t+1}) \mid S_t = s, Y_t^\eta = y, A_t = a \right], \tag{9}$$

*and*

$$\mathbb{E}^{\chi, \eta} \left[ -(\eta - R_{1:T})^+ \right] = \mathbb{E}_{A_1 \sim \chi_1(\cdot|s_1, \eta)} \left[ f_1^\chi(s_1, \eta, A_1) - g_1^\chi(s_1, \eta, A_1) \times \eta \right].$$

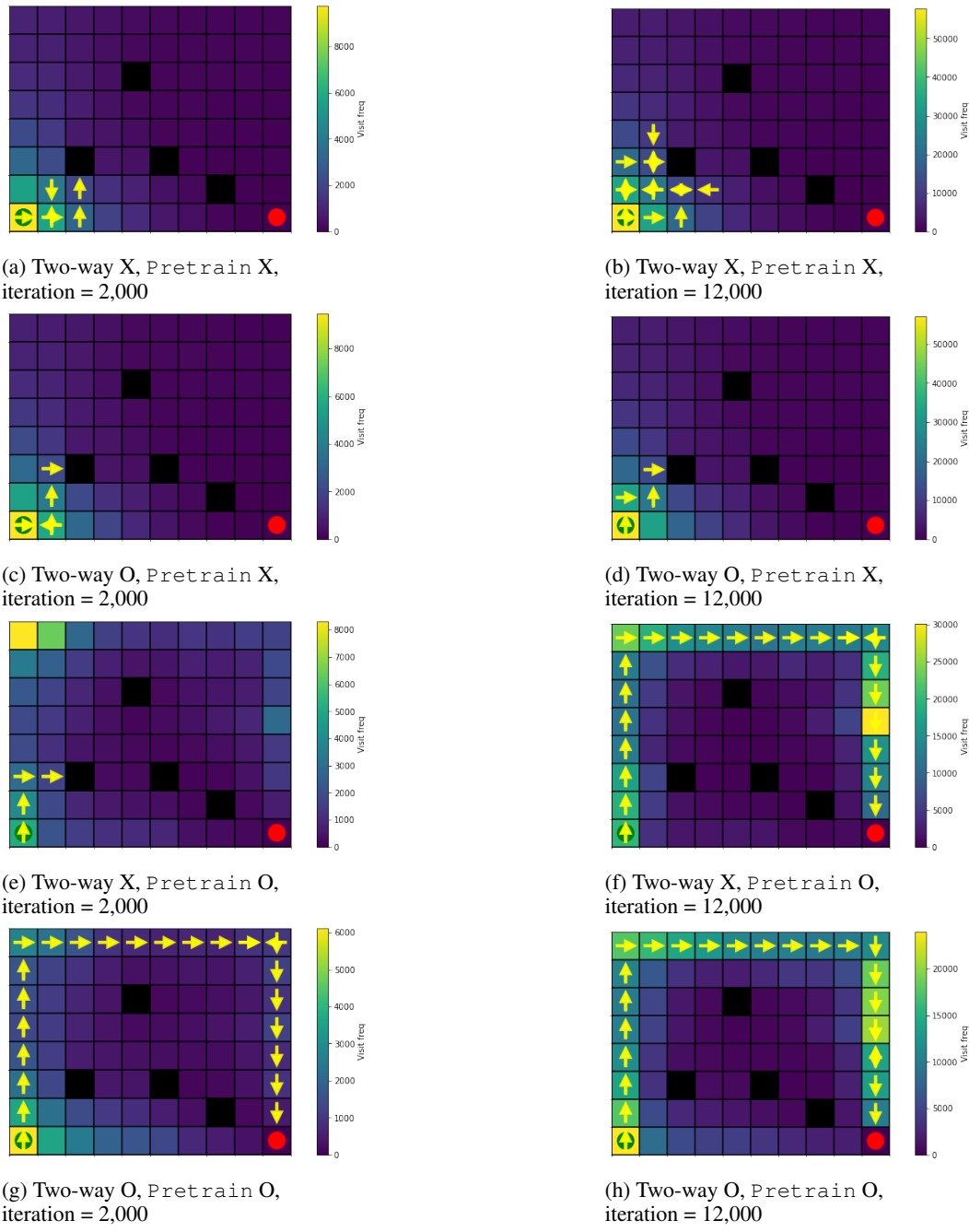

Figure 4: The distinct policies learned by PCVaR-Q agents over the course of Q-learning

*Proof.* The following results provide a key identity that allows us to express the event indicator over cumulative returns in terms of the predictive weight function $g^{\chi}$:

**Lemma 1.** *Given an augmented Markov policy kernel $\chi$ and risk budget $\eta$, under Assumption 1, we have*

$$\mathbb{E}^{\chi,\eta}[\mathbb{I}\{R_{t:T} \leq Y_t\}R_\tau | S_t, Y_t, A_t] = \mathbb{E}^{\chi,\eta}[g^{\chi}_{\tau+1}(S_{\tau+1}, Y_{\tau+1}, A_{\tau+1}) \times R_\tau | S_t, Y_t, A_t]$$

$\forall t \in \{1, \ldots, T\}$ *and* $\forall \tau \in \{t, \ldots, T\}$

*Proof.*

$$\mathbb{E}^{\chi,\eta}[\mathbb{I}\{R_{t:T} \le Y_t\}R_\tau|S_t, Y_t, A_t] \stackrel{(a)}{=} \mathbb{E}^{\chi,\eta}[\mathbb{I}\{R_{t:T} \le Y_t\}R_\tau|H_t, A_t]$$

$$\stackrel{(b)}{=} \mathbb{E}^{\chi,\eta}[\mathbb{E}^{\chi,\eta}[\mathbb{I}\{R_{t:T} \le Y_t\}R_\tau|H_{\tau+1}, A_{\tau+1}]|H_t, A_t]$$

$$\stackrel{(c)}{=} \mathbb{E}^{\chi,\eta}[\mathbb{E}^{\chi,\eta}[\mathbb{I}\{R_{t:T} \le Y_t\}|H_{\tau+1}, A_{\tau+1}]R_\tau|H_t, A_t]$$

$$= \mathbb{E}^{\chi,\eta}[\mathbb{E}^{\chi,\eta}[\mathbb{I}\{R_{\tau+1:T} \le Y_t - R_{t:\tau}\}|H_{\tau+1}, A_{\tau+1}]R_\tau|H_t, A_t]$$

$$\stackrel{(a)}{=} \mathbb{E}^{\chi,\eta}[\mathbb{E}^{\chi,\eta}[\mathbb{I}\{R_{\tau+1:T} \le Y_{\tau+1}\}|S_{\tau+1}, Y_{\tau+1}, A_{\tau+1}]R_\tau|H_t, A_t]$$

$$= \mathbb{E}^{\chi,\eta}[g_{\tau+1}^\chi(S_{\tau+1}, Y_{\tau+1}, A_{\tau+1}) \times R_\tau|H_t, A_t]$$

$$\stackrel{(a)}{=} \mathbb{E}^{\chi,\eta}[g_{\tau+1}^\chi(S_{\tau+1}, Y_{\tau+1}, A_{\tau+1}) \times R_\tau|S_t, Y_t, A_t]$$

We proceed by conditioning on the full history and applying the tower property of conditional expectation.

(a) By the Markov property with respect to the augmented state space, the conditional expectation given $(S_t, Y_t)$ is equivalent to conditioning on the full history $H_t$.

(b) We apply the law of total expectation to decompose the expectation across time steps, conditioning first on $H_{\tau+1}$.

(c) Since $H_{\tau+1}$ contains all rewards up to time $\tau$, we can isolate $R_\tau$ over the conditional expectation.

We utilize the definition of $g^\chi$ in the sixth step. Substituting this decomposition into the original expression yields the desired form. □

Applying Lemma 1, which is (d), to the definition of $f_t^\chi$, we obtain:

$$f_t^\chi(s, y, a) := \mathbb{E}^{\chi,\eta}[\mathbb{I}\{R_{t:T} \le Y_t\}R_{t:T}|S_t = s, Y_t = y, A_t = a]$$

$$= \mathbb{E}^{\chi,\eta}[\mathbb{I}\{R_{t:T} \le Y_t\}|S_t = s, Y_t = y, A_t = a]$$

$$\stackrel{(d)}{=} \mathbb{E}^{\chi,\eta}[\sum_{\tau=t}^T g_{\tau+1}^\chi(S_{\tau+1}, Y_{\tau+1}, A_{\tau+1}) \times R_\tau|S_t = s, Y_t = y, A_t = a]$$

Furthermore, similar technique utilizes the representation for $\mathbb{E}^{\chi,\eta}\left[-(\eta - R_{1:T})^+\right]$ as:

$$\mathbb{E}^{\chi,\eta}\left[-(\eta - R_{1:T})^+\right] = \mathbb{E}^{\chi,\eta}[(R_{1:T} - \eta) \times \mathbb{I}\{R_{1:T} \le Y_1\}]$$

$$= \mathbb{E}^{\chi,\eta}[R_{1:T} \times \mathbb{I}\{R_{1:T} \le Y_1\}] - \mathbb{E}^{\chi,\eta}[\eta \times \mathbb{I}\{R_{1:T} \le Y_1\}]$$

$$= \mathbb{E}^{\chi,\eta}[f_1^\chi(s_1, \eta, A_1)] - \mathbb{E}[g_1^\chi(s_1, \eta, A_1)] \times \eta$$

$$= \mathbb{E}_{A_1 \sim \chi_1(\cdot|s_1, \eta)}[f_1^\chi(s_1, \eta, A_1) - g_1^\chi(s_1, \eta, A_1) \times \eta].$$

Fot the last equation, we need the following lemma.

**Lemma 2.** *Given an augmented Markov policy kernel $\chi$ and risk budget $\eta$, under Assumption 1, we have*

$$\mathbb{E}^{\chi,\eta}[\mathbb{I}\{R_{t:T} \le Y_t\}|S_t, Y_t, A_t] = \mathbb{E}^{\chi,\eta}[g_{t+1}^\chi(S_{t+1}, Y_{t+1}, A_{t+1})|S_t, Y_t, A_t]$$

$\forall t \in \{1, \ldots, T\}$

*Proof.*

$$\mathbb{E}^{\chi,\eta}[\mathbb{I}\{R_{t:T} \le Y_t\}|S_t, Y_t, A_t] \overset{(a)}{=} \mathbb{E}^{\chi,\eta}[\mathbb{I}\{R_{t:T} \le Y_t\}|H_t, A_t]$$

$$\overset{(b)}{=} \mathbb{E}^{\chi,\eta}[\mathbb{E}^{\chi,\eta}[\mathbb{I}\{R_{t:T} \le Y_t\}|H_{t+1}, A_{t+1}]|H_t, A_t]$$

$$= \mathbb{E}^{\chi,\eta}[\mathbb{E}^{\chi,\eta}[\mathbb{I}\{R_{t+1:T} \le Y_t - R_t\}|H_{t+1}, A_{t+1}]|H_t, A_t]$$

$$\overset{(a)}{=} \mathbb{E}^{\chi,\eta}[\mathbb{E}^{\chi,\eta}[\mathbb{I}\{R_{t+1:T} \le Y_{t+1}\}|S_{t+1}, Y_{t+1}, A_{t+1}]|H_t, A_t]$$

$$= \mathbb{E}^{\chi,\eta}[g_{t+1}^{\chi}(S_{t+1}, Y_{t+1}, A_{t+1})|H_t, A_t]$$

$$\overset{(a)}{=} \mathbb{E}^{\chi,\eta}[g_{t+1}^{\chi}(S_{t+1}, Y_{t+1}, A_{t+1})|S_t, Y_t, A_t]$$

We proceed by conditioning on the full history and applying the tower property of conditional expectation.

(a) By the Markov property with respect to the augmented state space, the conditional expectation given $(S_t, Y_t)$ is equivalent to conditioning on the full history $H_t$.

(b) We apply the law of total expectation to decompose the expectation across time steps, conditioning first on $H_{\tau+1}$.

We utilize the definition of $g^{\chi}$ in the sixth step. Substituting this decomposition into the original expression yields the desired form. □

Applying Lemma 2, which is (d), to the definition of $g_t^{\chi}$, we obtain:

$$g_t^{\chi}(s, y, a) := \mathbb{P}^{\chi,\eta}\left( R_{t:T} \le y \mid S_t = s, Y_t^{\eta} = y, A_t = a \right)$$

$$= \mathbb{E}^{\chi,\eta}[\mathbb{I}\{R_{t:T} \le Y_t\}|S_t = s, Y_t = y, A_t = a]$$

$$\overset{(d)}{=} \mathbb{E}^{\chi,\eta}[g_{\tau+1}^{\chi}(S_{t+1}, Y_{t+1}, A_{t+1})|S_t = s, Y_t = y, A_t = a]$$

□

The first statement of Proposition 1 establishes a cumulative formulation of the predictive tail value function. This result naturally motivates a recursive Bellman-style decomposition. We formalize it in the next theorem.

## C.2 PROOF OF THEOREM 1

Building upon the temporal decomposition in Proposition 1, we derive a Bellman-type recursive relation for the predictive tail value function. This relation enables efficient policy evaluation and learning in dynamic settings.

**Theorem 1** (Bellman equation). *Given an augmented Markov policy kernel $\chi$, under Assumption 1, its predictive tail value function $f^{\chi}$ and predictive tail probability function $g^{\chi}$ satisfy*

$$f_t^{\chi}(s, y, a) = \mathbb{E}_{(R_t, S_{t+1}) \sim \mathcal{P}_t(\cdot|s,a), A_{t+1} \sim \chi_{t+1}(\cdot|S_{t+1}, y-R_t)}\big[f_{t+1}^{\chi}\big(S_{t+1}, y - R_t, A_{t+1}\big)$$
$$+ g_{t+1}^{\chi}\big(S_{t+1}, y - R_t, A_{t+1}\big) \times R_t\big],$$
$$\tag{10}$$

*for all $s \in \mathcal{S}$, $y \in \mathbb{R}$, $a \in \mathcal{A}$, and $t \in \{1, \ldots, T\}$.*

*Proof.*

$$f_t^\chi(s, y, a) = \mathbb{E}^{\chi, \eta} \left[ \sum_{\tau=t}^T g_{\tau+1}^\chi(S_{\tau+1}, Y_{\tau+1}, A_{\tau+1}) \times R_\tau \;\middle|\; S_t = s, Y_t^\eta = y, A_t = a \right]$$

$$= \mathbb{E}^{\chi, \eta} \left[ \sum_{\tau=t+1}^T g_{\tau+1}^\chi(S_{\tau+1}, Y_{\tau+1}, A_{\tau+1}) \times R_\tau \;\middle|\; S_t = s, Y_t^\eta = y, A_t = a \right]$$
$$+ \mathbb{E}^{\chi, \eta} \left[ g_{t+1}^\chi(S_{t+1}, Y_{t+1}, A_{t+1}) \times R_t \;\middle|\; S_t = s, Y_t^\eta = y, A_t = a \right]$$

$$= \mathbb{E}^{\chi, \eta} \left[ \sum_{\tau=t+1}^T g_{\tau+1}^\chi(S_{\tau+1}, Y_{\tau+1}, A_{\tau+1}) \times R_\tau \;\middle|\; S_t = s, Y_t^\eta = y, A_t = a \right]$$
$$+ \mathbb{E}_{(R_t, S_{t+1}) \sim \mathcal{P}_t(\cdot|s,a), A_{t+1} \sim \chi_{t+1}(\cdot|S_{t+1}, y - R_t)} \left[ g_{t+1}^\chi(S_{t+1}, y - R_t, A_{t+1}) \times R_t \right]$$
$$= \mathbb{E}^{\chi, \eta} \left[ f_{t+1}^\chi(S_{t+1}, y - R_t, A_{t+1}) \;\middle|\; S_t = s, Y_t^\eta = y, A_t = a \right]$$
$$+ \mathbb{E}_{(R_t, S_{t+1}) \sim \mathcal{P}_t(\cdot|s,a), A_{t+1} \sim \chi_{t+1}(\cdot|S_{t+1}, y - R_t)} \left[ g_{t+1}^\chi(S_{t+1}, y - R_t, A_{t+1}) \times R_t \right]$$
$$= \mathbb{E}_{(R_t, S_{t+1}) \sim \mathcal{P}_t(\cdot|s,a), A_{t+1} \sim \chi_{t+1}(\cdot|S_{t+1}, y - R_t)} \left[ f_{t+1}^\chi(S_{t+1}, y - R_t, A_{t+1}) \right.$$
$$\left. + g_{t+1}^\chi(S_{t+1}, y - R_t, A_{t+1}) \times R_t \right].$$

We begin with the temporal decomposition from Proposition 1, which expresses $f_t^\chi$ as a cumulative expectation of future rewards weighted by $g^\chi$:

$$f_t^\chi(s, y, a) = \mathbb{E}^{\chi, \eta} \left[ \sum_{\tau=t}^T g_{\tau+1}^\chi(S_{\tau+1}, Y_{\tau+1}, A_{\tau+1}) \cdot R_\tau \;\middle|\; S_t = s, Y_t = y, A_t = a \right].$$

We isolate the contribution of the first term $R_t$, and apply the law of total expectation with respect to the policy kernel $\chi$:

$$f_t^\chi(s, y, a) = \mathbb{E}_{(R_t, S_{t+1}) \sim \mathcal{P}_t(\cdot|s,a), A_{t+1} \sim \chi_{t+1}(\cdot|S_{t+1}, y - R_t)} \Big[$$
$$f_{t+1}^\chi(S_{t+1}, y - R_t, A_{t+1}) + g_{t+1}^\chi(S_{t+1}, y - R_t, A_{t+1}) \cdot R_t \Big].$$

This completes the recursive Bellman-type equation for $f^\chi$. $\qquad\square$

Having established the Bellman recursion for the predictive tail value function, we now turn to the corresponding optimality conditions. Theorem 2 characterizes the optimal value function and derives the form of the optimal policy via a greedy selection criterion.

### C.3 PROOF OF THEOREM 2

With the recursive formulation established in Theorem 1, we now turn to the optimality condition. The following result characterizes the optimal value function and policy structure under CVaR objective.

**Theorem 2** (Bellman optimality equation)**.** *Define*

$$v_t^\chi(s, y) := \mathbb{E}_{A_t \sim \chi_t(\cdot|s,y)} \left[ f_t^\chi(s, y, A_t) - g_t^\chi(s, y, A_t) \times y \right], \quad v_t^*(s, y) := \sup_{\chi \in \mathcal{X}} v_t^\chi(s, y).$$

*Then, the following holds under Assumption 1:*

1. *Let $\Pi(\chi)$ be the set of augmented Markov policies induced by a kernel $\chi$ across all values of $\eta \in \mathbb{R}$. Then,*

$$\sup_{\pi \in \Pi(\chi)} \left\{ q \cdot CVaR_q^\pi[R_{1:T}] \right\} = \max_{\eta \in \mathbb{R}} \left\{ q\eta + v_1^\chi(s_1, \eta) \right\}.$$

2. *With respect to all non-anticipating policies,*

$$\sup_{\pi \in \Pi} \left\{ q \cdot CVaR_q^\pi[R_{1:T}] \right\} = \max_{\eta \in \mathbb{R}} \left\{ q\eta + v_1^*(s_1, \eta) \right\}.$$

    3. $v^\chi \equiv v^*$ *if and only if $\chi$ is greedy with respect to $(f^\chi, g^\chi)$.*

*Proof.* For the first claim,
We begin by proving the first claim. For any fixed policy kernel $\chi$, we consider the class of augmented Markov policies $\Pi(\chi)$ it induces. Then,

$$
\sup_{\pi \in \Pi(\chi)} \left\{ q \cdot \mathrm{CVaR}_q^\pi[R_{1:T}] \right\} = \sup_{\pi \in \Pi(\chi)} \left\{ q \cdot \max_{\eta \in \mathbb{R}} \{ \eta + \mathbb{E}^\pi[-(\eta - R_{1:T})^+] \} \right\}
$$

$$
\overset{(a)}{=} \max_{\eta \in \mathbb{R}} \{ \eta q + \mathbb{E}^{\chi, \eta}[-(\eta - R_{1:T})^+] \}
$$

$$
\overset{(b)}{=} \max_{\eta \in \mathbb{R}} \{ \eta q + \mathbb{E}_{A_1 \sim \chi_1(\cdot | s_1, \eta)} [f_1^\chi(s_1, \eta, A_1) - g_1^\chi(s_1, \eta, A_1) \times \eta] \}
$$

$$
\overset{(c)}{=} \max_{\eta \in \mathbb{R}} \{ \eta q + v_1^\chi(s_1, \eta) \},
$$

where step (a) holds because the supremum over $\pi \in \Pi(\chi)$ includes the freedom to choose $\eta$, step (b) follows directly from Proposition 1 and step (c) follows directly from the definition of $v_t^\chi(s, y)$ in Theorem 2.
We now turn to the second claim. When optimizing over all admissible non-anticipating policies $\Pi$, we can equivalently optimize over the space of all Markov kernels $\chi \in \mathcal{X}$:

$$
\sup_{\pi \in \Pi} \left\{ q \cdot \mathrm{CVaR}_q^\pi[R_{1:T}] \right\} \overset{(a)}{=} \sup_{\chi \in \mathcal{X}} \left\{ \sup_{\pi \in \Pi(\chi)} \left\{ q \cdot \mathrm{CVaR}_q^\pi[R_{1:T}] \right\} \right\}
$$

$$
\overset{(b)}{=} \sup_{\chi \in \mathcal{X}} \left\{ \max_{\eta \in \mathbb{R}} \{ q\eta + v_1^\chi(s_1, \eta) \} \right\}
$$

$$
= \max_{\eta \in \mathbb{R}} \left\{ q\eta + \sup_{\chi \in \mathcal{X}} \{ v_1^\chi(s_1, \eta) \} \right\}
$$

$$
\overset{(c)}{=} \max_{\eta \in \mathbb{R}} \{ q\eta + v_1^*(s_1, \eta) \},
$$

where step (a) uses the policy optimization result (Theorem 3.2 in (Bäuerle & Ott, 2011)), step (b) follows from the result already shown in the first claim of this theorem, and step (c) uses the definition of $v_t^*(s, y)$ as the pointwise supremum over $\chi \in \mathcal{X}$.
To prove the final statement, we make use of the optimality condition for value functions (Theorem 5.1 in (Wang et al., 2023)), which states that equality between $v^\chi$ and $v^*$ holds if and only if the policy is greedy with respect to the action-value function $u^\chi$. Specifically:

$$
v^\chi \equiv v^* \Leftrightarrow \left( \{ a \in \mathcal{A} \mid \chi_t(a | s, y) > 0 \} \subseteq \arg\max_{a \in \mathcal{A}} u_t^\chi(s, y, a) \quad \forall t, s, y. \right)
$$

Using the identity $u_t^\chi(s, y, a) = f_t^\chi(s, y, a) - g_t^\chi(s, y, a) \cdot y$, the equivalence follows.

$\square$

## C.4   Proof of theorem 3

We now show that a policy improvement guarantee in the context of CVaR optimization over augmented state spaces. This forms the backbone of our Predictive CVaR Q-learning algorithms in risk-sensitive reinforcement learning settings.

**Theorem 3** (Policy improvement). *Consider an augmented Markov policy kernel $\chi$ along with its predictive tail value function $f^\chi$ and predictive tail probability function $g^\chi$. Let $\chi'$ be the greedy kernel with respect to $(f^\chi, g^\chi)$. Then, under Assumption 1,*

$$
v_t^\chi(s, y) \leq v_t^{\chi'}(s, y), \quad \forall s \in \mathcal{S}, y \in \mathbb{R}, t \in \{1, \ldots, T\}.
$$

*Consequently,*

$$
\sup_{\pi \in \Pi(\chi)} CVaR_q^\pi(R_{1:T}) \leq \sup_{\pi \in \Pi(\chi')} CVaR_q^\pi(R_{1:T}), \tag{11}
$$

*for any $q \in (0, 1]$.*

*Proof.*

$$\sup_{\pi \in \Pi(\chi)} \text{CVaR}_q^\pi(R_{1:T}) \stackrel{(a)}{=} \max_{\eta \in \mathbb{R}} \{q\eta + v_1^\chi(s_1, \eta)\}$$

$$\stackrel{(b)}{\leq} \max_{\eta \in \mathbb{R}} \left\{ q\eta + v_1^{\chi'}(s_1, \eta) \right\}$$

$$\stackrel{(a)}{=} \sup_{\pi \in \Pi(\chi')} \text{CVaR}_q^\pi(R_{1:T})$$

where step (a) uses *1.* of Theorem 2, step (b) follows from the definition of greedy kernel $\chi'$. $\quad\square$

