# OpenReview forum: "Predictive CVaR Q-learning"
_ICLR.cc/2026/Conference — ICLR 2026 Poster_

### Official Review · Reviewer_wLib · 2025-10-27

**Soundness:** 2
**Presentation:** 3
**Contribution:** 3
**Rating:** 2
**Confidence:** 4

**Summary:**

The authors propose a novel CVaR recursive structure based on predictive tail value functions and predictive tail probability functions. This recursive structure, along with a risk-sensitivity exploration strategy, leads to superior empirical performance when compared to the usual CVaR recursive structure.

**Strengths:**

The proposed CVaR recursive structure based on predictive tail value functions and predictive tail probability functions is innovative and clever. The theoretical work done to support this recursive structure is sound (albeit with some typos; see weaknesses). The empirical results are encouraging.

**Weaknesses:**

Overall, I would consider this paper to be sound work. However, I do have some concerns that would need to be addressed prior to publication.

My biggest concern is that the work performed does not validate the authors’ claims. In particular, the authors mention numerous times that the primary benefit of their proposed method is sample efficiency, yet there is no theoretical work done in support of this claim, and the empirical results do not provide any evidence of such claims. The closest measure of sample efficiency that can be made from the empirical results is in Figures 2/3b), however in Figure 2b), none of the methods converge to the same solution so a proper evaluation of sample efficiency cannot be made. In Figure 3b), there is no statistical difference between the methods.

Moreover, I am not convinced (i.e., it was not rigorously argued by the authors) that value-based CVaR methods have the same sample efficiency issues that policy-gradient methods do. In particular, the claim that value-based methods lead to significant sample inefficiency in lines 165-166 is not proved by the authors, nor do they provide a citation to back up these claims. In fact, a key piece of theoretical work that would greatly enhance the paper is some result that shows why the regular CVaR decomposition (Equation 2) is sample inefficient.

Another concern is that the empirical analysis lacks focus. In particular, in addition to the concerns mentioned above, the use of function approximation for the simple experiments considered in this paper seems unnecessary. I would argue that if the authors want to include the work related to function approximation, they need to provide a compelling experiment that makes proper use of it. Furthermore, it seems odd that for the simple experiments included in the paper, that neither method can find the optimal solution.

It is also not clear whether the shown gain in performance is entirely due to the risk-sensitivity exploration strategy. In particular, was the same exploration strategy used with the baseline algorithm as well? If so, then why even include it in the first place when making the comparison (i.e., would it not be a cleaner comparison without the exploration strategy)? If not, then this is not a proper evaluation of the proposed algorithm's performance, and an ablation study would be needed.

Overall, although I see a lot of merit in the work performed by the authors, the current draft of the paper makes it seem like a ‘forced’ adaptation of prior work done in the policy-gradient domain into the value-based domain, rather than a purposeful, adequately-motivated endeavour.

Accordingly, in order to increase my score, the authors would need to: 1a) provide theoretical and/or empirical results that support their claims related to sample efficiency, or 1b) remove the claims of sample efficiency and find a more compelling narrative for the paper. 2) The authors would also need to address my concerns related to the empirical analysis.

**Minor Comments:**
- The introduction is unfocused and hard to read. In particular, the constant switching between policy-gradient and value-based methods is hard to follow. Overall, I do not see a reason to mention policy-gradient methods at all in this paper.
- Line 31: I would argue that CVaR has a lot of tractability issues (which is why it is such a difficult objective to optimize) and that the primary reason that it is valued is because it is a coherent risk measure.
- Lines 46-51: The discussion in this paragraph completely ignores the notion of dynamic risk measures, which would need to be mentioned to make a proper argument.
- Section 2 would greatly benefit by having more citations related to the methods that the authors are building upon.
- Appendix C needs equation numbers.
- Lemma 2 is filled with several copy/paste errors from Lemma 1 (e.g. c) is not needed in this proof)

**Questions:**

Lines 183-184: can the authors expand on why the choice of $\eta$ is arbitrary? This seems counterintuitive to me.

---

> ### Comment · Area_Chair_mmra · 2025-11-13
> **About the weakness**
>
> Dear reviewer wLib,
>
> The Openreview system seems to have an error. Your comment on Weakness is the same as that of reviewer Dyty. If this is not your comment. Could you please try to make an edit and update this part. If this does not work, could you please post your report of weakness as a comment under your report?
>
> Best, AC

---

### Official Review · Reviewer_Dyty · 2025-10-29

**Soundness:** 3
**Presentation:** 3
**Contribution:** 3
**Rating:** 8
**Confidence:** 3

**Summary:**

The paper addresses the challenges in risk-sensitive reinforcement learning (RL) using the Conditional Value-at-Risk (CVaR) objective, which focuses on optimizing the expected return in the worst-case quantile of the return distribution (e.g., for safety-critical applications like autonomous driving or finance). Standard CVaR RL methods are sample-inefficient due to two key issues: (1) noisy policy evaluation from treating CVaR as a non-decomposable, terminal objective, which delays learning signals and hinders temporal credit assignment; and (2) "blindness to success," where the agent ignores high-return trajectories outside the risk tail, leading to premature convergence to overly conservative, suboptimal policies.

**Strengths:**

1. The paper proposed CVaR recursive structure based on predictive tail value functions and predictive tail probability functions is innovative and clever.
2. The theoretical work done to support this recursive structure is sound (albeit with some typos; see weaknesses).
3. The empirical results are encouraging.

**Weaknesses:**

Overall I would consider this paper to be sound work. However, I do have some concerns that would need to be addressed prior to publication.

My biggest concern is that the work performed does not validate the authors’ claims. In particular, the authors mention numerous times that the primary benefit of their proposed method is sample efficiency, yet there is no theoretical work done in support of this claim, and the empirical results provided do not provide any evidence of such claims. The closest measure of sample efficiency that can be made from the empirical results is in Figures 2-(b) and  3-(b), however in Figure 2-(b), none of the methods converge to the same solution so a proper evaluation of sample efficiency cannot be made. In Figure 3-(b), there is no statistical difference between the methods.

Moreover, I am not convinced (i.e., it was not rigorously argued by the authors) that value-based CVaR methods have sample efficiency issues in the same way that policy gradient methods do. In particular, the claim that value-based methods lead to significant sample inefficiency in lines 165-166 is not proved by the authors, nor do they provide a citation to back up these claims. In fact, a key piece of theoretical work that would greatly enhance the paper is some results that show why the regular CVaR decomposition (Equation 2) is sample inefficient.

Another concern is that the empirical analysis lacks focus. In particular, in addition to the concerns mentioned above, the use of function approximation for the simple experiments considered in this paper seems unnecessary. I would argue that if the authors want to include the work related to function approximation, they need to provide a compelling experiment that makes proper use of it. Furthermore, it seems odd that for the simple experiments included in the paper, neither method can find the optimal solution.

It is also not clear whether the shown gain in performance is due to the risk-sensitivity exploration strategy. In particular, was the same exploration strategy used with the baseline algorithm as well? If so, then why even include it in the first place when making the comparison (i.e., would it not be a cleaner comparison without the exploration strategy)? If not, then this is not a proper evaluation of the proposed algorithm's performance.

Overall, although I see a lot of merit in the work performed by the authors, the current draft of the paper makes it seem like a ‘forced’ adaptation of prior work done in the policy-gradient domain into the value-based domain, rather than a purposeful, adequately-motivated endeavor.

Accordingly, in order to increase my score, the authors would need to: 1a) provide theoretical and/or empirical results that support their claims related to sample efficiency, or 1b) remove the claims of sample efficiency and find a more compelling narrative for the paper. The authors would also need to address my concerns related to the empirical analysis.

**Questions:**

Lines 183-184: can the authors expand on why the choice of $\eta$ is arbitrary? This seems counterintuitive to me.

---

> ### Comment · Area_Chair_mmra · 2025-11-13
> **About the weakness**
>
> Dear reviewer Dyty,
>
> The Openreview system seems to have an error. Your comment on Weakness is the same as that of reviewer wLib. If this is not your comment. Could you please try to make an edit and update this part. If this does not work, could you please post your report of weakness as a comment under your report?
>
> Best,
> AC

---

### Official Review · Reviewer_9ktc · 2025-10-31

**Soundness:** 3
**Presentation:** 3
**Contribution:** 3
**Rating:** 6
**Confidence:** 3

**Summary:**

This paper proposes a sample-efficient Q-learning algorithm (PCVaR-Q) to optimize the Conditional Value-at-Risk (CVaR) target. Its core contribution is two key innovations: First, the "predictive tail value function" is proposed, which constructs a novel recursive structure suitable for CVaR targets, similar to traditional risks. The neutral Bellman equation aims to solve the problem of noise strategy evaluation caused by the indecomposition of the target. Second, introduce a "two-way exploration" strategy, which explores the risk sensitivity of action space and intelligent bodies at the same time, so as to alleviate the "blindness to success" phenomenon.

**Strengths:**

1. The core innovations lie in the proposal of the predictive tail value function $f^\chi$ and the predictive tail probability function $g^\chi$ for policy kenel $\chi$. Based on the newly defined tail functions, Theorem 1 & 2 gives the Bellman equation and the Bellman optimality equation.
2. Combined with the newly developed tail value function and probability function, the proposed "two-way randomized exploration" approach explicitly solves the known problem of "blindness to success" in CVaR learning, and encourages intelligent bodies to explore strategies with different risk preferences. This is achieved by samplingys around the risk budget $\eta$.
3. The experimental results (Figures 2 and 3) strongly support the author's argument. Compared with the CVaR-Q baseline, PCVaR-Q shows higher stability and lower variance during the training process.

**Weaknesses:**

1. The entire theoretical framework (especially Theorem 1 & 2) depends on Assumption 1, which states that the distribution of the residual return $R_{t:T}$ has no probability mass. However, many standard reinforcement learning environments (including discrete rewards or deterministic rewards) would violate this assumption. Moreover, I found that the experiements environment considered (such as the. Sequential decision tree setting) clearly violates Assumption 1. The author did not discuss the impact of the violation of thies assumption.
2. TThe pre-training step is crucial for success, as all models that did not undergo pre-training (Figure 4a-d) failed to learn the optimal path. This seems to weaken the argument that the algorithm has a robust exploration strategy and "sample-efficient," as exploration seems to fail in cold-start case.
3. Currently, the experiments in the main text (Section 5) have failed to clearly disentangle the contributions of the three main contributions—(1) the new Bellman equation, (2) pre-training, and (3) two-way random exploration—to the performance improvement.

**Questions:**

1. For the augmented state $(s, y, a)$, how is the "table-based function approximator" implemented? Considering that $y$ is a continuous variable, is discretization based on the grid $H$ used?
2. Without using pre-training, to what extent can your proposed new Bellman equation (Eq. 5) and bidirectional exploration itself solve the issues of "blindness" and learning instability?


---
Typos

1. Line 369, "through risk-nuetral Q-learning.": nuetral -> neutral.
1. Line 441, "experimental environment and and the distinct policies": there are two "and" here.
2. Line 472, "a novel novel CVaR Q-learning framework": double "novel".

---

> ### Author Response · Authors · 2025-11-18
>
> Thank you for this thoughtful and insightful comment. We also note and will correct the minor typo in the original manuscript.
>
> **Assumption 1 and its violation**: We thank the reviewer for this insightful comment. We would like to clarify that Assumption 1 is not directly used in the proofs; rather, it is introduced to ensure the continuity and smoothness of ($f$) and ($g$), as well as the uniqueness of the optimal ($\eta^*$), which helps make the theoretical derivations cleaner. We also consider this assumption useful when extending to DQN and computing gradients. In practice, even in environments with discrete rewards (i.e., sequential decision tree), our algorithm operates correctly. Gradients can still be computed using subgradients or reward smoothing techniques, allowing learning despite the violation of this assumption.
>
> **Sample-efficiency and robust exploration**: We appreciate the reviewer’s sharp observation. We agree that, in the grid-world experiment, pre-training played a significant role in achieving successful learning, and we note that the baseline models also succeeded only when pre-trained. The empirical results indeed indicate that risk-level exploration alone does not fully resolve the “blindness to success” issue in this particular environment. However, regarding sample efficiency, we believe that the comparison with the baseline and the results in Figure 4(e–f, g–h) demonstrate meaningful improvements.
>
> In terms of how each component contributes to addressing the “blindness to success” issue (robust exploration), our view is that risk-level exploration broadens the search over estimated risk levels, while pre-training helps obtain a more accurate initial estimate of the risk level. In the grid-world setting, random initial exploration often yields risk-level estimates that are far from the target risk level; consequently, exploration around such inaccurate estimates provides limited infomation. Thus, pre-training becomes disproportionately important in this environment. Despite this limitation, we included the grid-world experiment because it is a canonical benchmark for evaluating tabular Q-learning–style methods.
>
> **Disentangling contributions in Section 5**: We appreciate the reviewer’s concern regarding the separation of the contributions of (1) the new Bellman equation, (2) pre-training, and (3) bidirectional exploration.
>
> Our intention in Section 5 was to highlight the effect of the new Bellman equation through comparisons with existing baselines. However, we agree that these experiments do not show holding new Bellman equation as directly as they could. To address this, we are considering an additional diagnostic experiment comparing heatmaps of ($f_{t+1}$) and ($r_t \cdot g_t + f_t$), which would more explicitly illustrate the behavior induced by the new Bellman equation . We thank the reviewer for prompting this clarification. We will make this purpose explicit in the revised main text.
>
> While the contributions of pre-training and bidirectional exploration were presented in the appendix due to page limitations, we plan to integrate these ablation results into the main body of the paper in the revised version.
>
> **Table-based function approximator for continuous $y$**: Exactly as the reviewer inferred, we discretize the continuous variable ($y$) using the grid ($H$). In the MDP experiments, ($H$) spans the range ([-10, 15]) with increments of 0.1, and in the grid-world experiments, it spans ([-150, 100]) with increments of 1.0. This discretization is then used for the tabular approximation of our predictive tail functions. We will clarify this implementation detail more explicitly in the revised version.
>
> **Effectiveness of the new Bellman equation and bidirectional exploration without pre-training**: The new Bellman equation reduces evaluation error and improves stability by decomposing the predictive tail probability and predictive tail value, thereby mitigating the instability issues that arise in classical CVaR Q-learning.
>
> Regarding the “blindness” issue, our view is that bidirectional (risk-level) exploration broadens the search over the estimated risk levels, while pre-training provides a more accurate initial estimate of the relevant risk region. In the grid-world experiments, the environment places substantial emphasis on obtaining a good initial estimate of the risk level, because performing random exploration at the beginning often leads to an unreasonable initial estimate of the risk level (e.g., close to the negative maximum episode length), resulting in very few trajectories corresponding to that risk level. In contrast, in the sequential decision tree setting the risks associated with success and failure were more similar, making broad risk-level exploration particularly beneficial even without pre-training.
>
> We will clarify these distinctions and the complementary roles of the two components in the revised version.

---

### Official Review · Reviewer_fM76 · 2025-11-01

**Soundness:** 3
**Presentation:** 3
**Contribution:** 3
**Rating:** 6
**Confidence:** 3

**Summary:**

The paper introduces Predictive CVaR Q-Learning (PCVaR-Q), which makes CVaR optimization TD-learnable by defining predictive tail value and predictive tail probability functions that satisfy a new Bellman recursion. This enables step-wise CVaR learning and supports a proven policy-improvement guarantee.

**Strengths:**

- Clear theoretical innovation: a Bellman-consistent, value-based formulation for CVaR.
- Strong proofs and solid connection to policy improvement.
- Well-written and conceptually clear.

**Weaknesses:**

- Limited empirical scope: Experiments are small-scale and tabular; results demonstrate feasibility but not scalability.
- Comparison gaps: The paper could benchmark against more recent risk-sensitive or distributional RL algorithms (e.g., D4PG, IQN with tail weighting).
- Exploration heuristic: The “risk-level exploration” scheme is sensible but empirically underexplored.

**Questions:**

1. How sensitive is learning stability to the sampling of risk levels during exploration?
2. Can the predictive-tail recursion extend naturally to actor–critic or deep function-approximation settings?
3. Does the algorithm handle non-stationary return distributions or stochastic environments robustly?

---

> ### Author Response · Authors · 2025-11-18
>
> Thank you for your invaluable comments.
>
> **Limited empirical scope**: We agree that the current experiments are small-scale and tabular, but the grid-world domain was chosen as a representative and illustrative example for Q-learning. In addition, we expect that our algorithm can scale to larger problems using deep function approximators such as DQN. In fact, we are currently conducting experiments on a driving game [1] using DQN, and we will make every effort to incorporate these results into the final version of the paper.
>
> **Comparison gaps**: We appreciate the reviewer’s sharp observation. While we discussed related CVaR policy-gradient and distributional RL approaches in the paper, our experimental focus was to isolate and highlight the effect of our reformulation and temporal decomposition. For this reason, we used the classical CVaR Q-learning algorithm as the primary baseline. As our DQN-based implementation nears completion, we plan to incorporate additional baselines—including those suggested by the reviewer—. We thank the reviewer for pointing out these valuable candidates.
>
> **Exploration heuristic**: We agree that the proposed risk-level exploration mechanism is empirical in nature. Similar to ε-greedy exploration, its exploration is governed by two design choices: the initial value of σ and the schedule by which σ decays over time. The initial σ can be reasonably specified from prior domain knowledge or from basic properties of the environment. The decay schedule, much like the choice of ε-decay in ε-greedy, admits multiple reasonable forms and is ultimately selected based on empirical validation.
>
> Our approach follows this established practice: risk-level exploration provides a flexible heuristic whose hyperparameters can be tuned through standard experimental procedures, just as is common for classical ε-greedy exploration strategies. We will clarify this point in the revision.
>
> **Sensitivity to sampling of risk levels**: We have experimentally evaluated the method under several choices of initial \sigma values and multiple decay schedules. Across these variations, we observed that the learning stability is not significantly affected. This suggests that, in practice, the sampling strategy for risk levels can be configured reliably through modest empirical tuning.
>
> **Extension to deep function approximation or actor–critic**: We believe that the predictive-tail approach can be integrated with deep function approximation, such as DQN, which we are currently exploring. It may also be combined with predictive CVaR policy-gradient methods [2] to naturally extend to actor–critic frameworks. However, additional care is required when propagating predictive tail probabilities in such settings, because—unlike in predictive CVaR policy-gradient methods—our definitions of the predictive tail value and probability functions are explicitly action-conditioned. This action dependence calls for careful treatment when designing the propagation mechanism, particularly within the temporal structure of deep RL architectures.
>
> **Handling of stochastic returns or non-stationary**: We thank the reviewer for raising this important point. Regarding stochastic environments, we note that CVaR-based objectives become more meaningful precisely when the environment exhibits stochasticity, as risk-sensitive criteria capture behaviors that are not reflected by expectation alone. Our analysis already assumes a stochastic MDP setting, so our formulation naturally accommodates stochastic environments.
>
> Non-stationary return distributions, however, fall outside the assumptions used in our analysis. Extending our predictive-tail formulation to such settings would require defining an adaptive augmented Markov policy kernel that explicitly account for time-varying reward distributions. We believe this is a promising and feasible direction for our algorithm: if one can model or estimate the non-stationarity in the reward process, then a time-indexed or adaptive predictive-tail kernel could naturally generalize our framework. Moreover, because our algorithm employs risk-level exploration, we expect it to adapt more robustly to distributional shifts (e.g., changes in variance or tail behavior) than methods based solely on fixed risk levels.
>
> We appreciate the reviewer for highlighting this valuable research direction.
>
> **References**
>
> [1] Greenberg, Ido, et al. "Efficient risk-averse reinforcement learning." Advances in Neural Information Processing Systems 35 (2022): 32639-32652.
>
> [2] Ju-Hyun Kim and Seungki Min. Risk-sensitive policy optimization via predictive cvar policy gradient.

---

### Official Review · Reviewer_FfEG · 2025-11-16

**Soundness:** 2
**Presentation:** 3
**Contribution:** 2
**Rating:** 4
**Confidence:** 3

**Summary:**

This paper proposes a new method for RL with a Conditional Value-at-Risk objective. To address the challenges posed by nonlinearity and non-decomposability, the authors introduce predictive value/probability functions and develop a new RL algorithm based on them.

**Strengths:**

The motivation and mathematical development are clearly presented. While the derivations seem fairly standard, they provide a clear motivation for the proposed algorithm.

**Weaknesses:**

The paper's contribution requires further justification.

1. After formulating objective (1), one could straightforwardly apply an actor–critic approach to optimize $ \mathbb{E}^{\chi,\eta}[-(\eta - R_{1:T})^{+}]$ (e.g., REINFORCE). It is not evident that the proposed method is superior to such actor-critic methods. While it may be true that only trajectories with non-zero effective reward are informative for actor-critic, the same limitation appears to affect the proposed algorithm: since g models the tail probability, when only a small subset of trajectories has non-zero effective reward, the estimate of g is likely to be noisy.

2. The experiments are limited to toy settings and do not include comparisons against actor-critic methods or stronger baselines. A more comprehensive comparison would also address the weakness noted above.

**Questions:**

See above.

---

> ### Author Response · Authors · 2025-11-18
>
> Thank you for this thoughtful and insightful comment.
>
> **Superiority of our method and estimation noise of $g$**: We thank the reviewer for this sharp and insightful comment. We agree that, just as the estimation of ( $\mathbb{E}[-(\eta - R_{1:T})^+] $) can be noisy when only a small subset of trajectories yields non-zero effective reward, the estimation of ($g$) can similarly suffer from noise under such sparsity.
>
> However, we believe that estimating a probability ($g$) rather than a tail expectation ($\mathbb{E}[-(\eta - R_{1:T})^+] $) offers an important advantage in these sparse-signal regimes. In particular, this advantage becomes most pronounced in the challenging cases (1) where the number of trajectories with non-zero effective reward is small, and (2) where the rewards beyond the threshold exhibit large variance. The tail probability ($g$) can still be estimated in a relatively stable way—e.g., by counting the proportion of trajectories exceeding the threshold. In contrast, estimating ($ \mathbb{E}[-(\eta - R_{1:T})^+] $) requires regressing on the magnitude of the tail values, which becomes considerably more unstable when the effective-reward samples are both few and high-variance.
>
> Our method explicitly separates probability estimation (via ($g$)) and value estimation (via ($f$)) through the temporal decomposition we introduce. This decoupling allows the probability component to be learned in a more stable manner, while the value component can leverage the decomposition structure to reduce variance. Moreover, since ($g$) represents a probability bounded in ([0,1]), it can benefit from stable learning strategies, as the reviewer noted, such as log-loss or KL-divergence based objectives, further mitigating noise.
>
> We will clarify these points in the revision and explicitly discuss the comparative advantages and limitations relative to actor–critic approaches.
>
> **Limited experiments and comparisons**: We appreciate the reviewer’s insightful comments regarding the experimental scope. We agree that the current experiments are small-scale and tabular. The grid-world domain was chosen as a representative setting to cleanly isolate and highlight the effect of our reformulation and temporal decomposition, and thus we used the classical CVaR Q-learning algorithm as the primary baseline. That said, we concur that more comprehensive comparisons—including actor-critic methods and stronger baselines—would strengthen the empirical evaluation and help address the reviewer’s concerns.
>
> To address experimental concern, we are actively extending our study beyond the tabular setting. In particular, we are implementing a deep version of our method using DQN, which allows us to evaluate performance in larger and more realistic domains. We are currently running experiments on a Driving game [1], and we will make every effort to include these results in the final version. As this implementation nears completion, we also plan to incorporate additional baselines (e.g., actor-critic and distributional RL methods) to provide a more extensive comparison.
>
> We thank the reviewer again for this helpful suggestion and will update the paper accordingly.
>
> **Reference**
>
> [1] Greenberg, Ido, et al. "Efficient risk-averse reinforcement learning." Advances in Neural Information Processing Systems 35 (2022): 32639-32652.

---

### Meta-Review · Area_Chair_AMg6 · 2025-12-11

**Summary:**

Reviewer wLib and Byty have very strange behavior, and thus we ignore them.

For the remaining reviewers, the strengths highlighted include:

i) Clear motivation and theoretical innovation for the proposed algorithm

ii) Strong proofs and solid connection to policy improvement

iii) Clear writting

The main weaknesses concern the experimental section:

i) Experiments are limited to toy examples.

ii) Lack of recent baselines.

iii) The ablations related to "pre-training" and "two-way random exploration" are not sufficiently explained.

Additionally, one reviewer comments that the proposed method may appear somewhat straightforward.

**Reviewer Concerns:**

In my opinion, the comment that the method may appear straightforward is valid but not quite critical, and the authors have addressed this point to some extent in the rebuttal. The more substantial concerns relate to the experimental evaluation. In the rebuttal, the authors said that they will include additional experiments, covering more realistic problem instances as well as more recent baselines. Assuming the results remain consistent, I consider this concern to be partially addressed.

**Reviewer Scores:**

Reviewer FfEG may increase the score by 1, or may not. It is hard to say.

Reviewer fM76 and 9ktc will keep their scores.

---

### Decision · Program_Chairs · 2026-01-26

Accept (Poster)